# The *Kras^G12D;Trp53^fl/fl* murine model of undifferentiated pleomorphic sarcoma is macrophage dense, lymphocyte poor, and resistant to immune checkpoint blockade

**Karys M. Hildebrand**[1,2,3], **Arvind K. Singla**[1,2,3], **Reid McNeil**[3,4], **Kayla L. Marritt**[1,2,3], **Kurt N. Hildebrand**[1,2,3], **Franz Zemp**[3,4,5], **Jahanara Rajwani**[3,4,5], **Doha Itani**[6], **Pinaki Bose**[3,4], **Douglas J. Mahoney**[3,4,5], **Frank R. Jirik**[2,5,7], **Michael J. Monument**[1,2,3]*

1 Department of Surgery, Cumming School of Medicine, University of Calgary, Calgary, Alberta, Canada,
2 McCaig Bone and Joint Institute, Cumming School of Medicine, University of Calgary, Calgary, Alberta, Canada, 3 Arnie Charbonneau Cancer Research Institute, Cumming School of Medicine, University of Calgary, Calgary, Alberta, Canada, 4 Department of Microbiology, Immunology and Infectious Diseases, Cumming School of Medicine, University of Calgary, Calgary, Alberta, Canada, 5 Alberta Children's Hospital Research Institute, University of Calgary, Calgary, Alberta, Canada, 6 Department of Pathology and Laboratory Medicine, Medical College of Wisconsin, Milwaukee, WI, United States of America, 7 Department of Biochemistry and Molecular Biology, Cumming School of Medicine, University of Calgary, Calgary, Alberta, Canada

* mjmonume@ucalgary.ca

## Abstract

Sarcomas are rare, difficult to treat, mesenchymal lineage tumours that affect children and adults. Immunologically-based therapies have improved outcomes for numerous adult cancers, however, these therapeutic strategies have been minimally effective in sarcoma so far. Clinically relevant, immunologically-competent, and transplantable pre-clinical sarcoma models are essential to advance sarcoma immunology research. Herein we show that Cre-mediated activation of *Kras^G12D*, and deletion of *Trp53*, in the hindlimb muscles of C57Bl/6 mice results in the highly penetrant, rapid onset undifferentiated pleomorphic sarcomas (UPS), one of the most common human sarcoma subtypes. Cell lines derived from spontaneous UPS tumours can be reproducibly transplanted into the hindlimbs or lungs of naïve, immune competent syngeneic mice. Immunological characterization of both spontaneous and transplanted UPS tumours demonstrates an immunologically-'quiescent' microenvironment, characterized by a paucity of lymphocytes, limited spontaneous adaptive immune pathways, and dense macrophage infiltrates. Macrophages are the dominant immune population in both spontaneous and transplanted UPS tumours, although compared to spontaneous tumours, transplanted tumours demonstrate increased spontaneous lymphocytic infiltrates. The growth of transplanted UPS tumours is unaffected by host lymphocyte deficiency, and despite strong expression of PD-1 on tumour infiltrating lymphocytes, tumours are resistant to immunological checkpoint blockade. This spontaneous and transplantable immune competent UPS model will be an important experimental tool in the pre-clinical development and evaluation of novel immunotherapeutic approaches for immunologically cold soft tissue sarcomas.

**Data Availability Statement:** RNAseq data files can be found in the NCBI Gene Expression Omnibus (accession number GSE174540).

**Funding:** This work was supported by funding from the Alberta Cancer Foundation, Alberta Children's Hospital Foundation, and the Cancer Research Society, all to MJM. The funders had no role in study design, data collection and analysis, decision to publish, or preparation of the manuscript.

**Competing interests:** The authors have declared that there are no competing interests that exist.

## Introduction

Sarcomas are connective tissue-derived malignancies that affect individuals of all ages and occur in any anatomical site of mesodermal origin [1]. Relative to carcinomas and hematologic malignancies, sarcomas are rare, yet they account for 15–20% of solid cancers in children and young adults [2]. Soft tissue sarcomas (STS), originate in skeletal muscle, vascular smooth muscle, adipose tissue, and fibroblastic mesenchymal progenitor cells. As a general rule, STS are highly resistant to systemic therapies [3]. Therapeutic advancements for STS have been stagnant for decades, and treatment options for STS patients with unresectable metastatic disease are extremely limited [4, 5].

Immunologically-based therapies have improved outcomes for numerous adult solid cancers [6–9]. Unfortunately, these immunotherapy benefits have largely not been observed in STS. Recent clinical trials have consistently failed to show significant clinical improvements for advanced STS patients treated with immune checkpoint inhibitors [10–12], as well as other immune-based strategies including systemic administration of cytokines or chemokines [13–15], vaccines [16–18] and adoptive cell transfers [19]. There are over 50 subtypes of STS, however, a recent study used genomic data to classify human sarcomas based on the tumour immune microenvironment (TIME), defining these as either immunologically cold, vascular, and lymphocyte high [20]. Immunologically cold, the most common STS category, predicted poor clinical response to immune checkpoint blockade. Other groups have similarly documented immunologically 'cold' as the dominant sarcoma phenotype, is characterized by a paucity of tumour lymphocytes [21–23] and a macrophage-rich, immune quiescent TIME [24–28]. Recent clinical data from over 1200 STS patients reaffirms the rich, predominantly pro-tumorigenic M2 infiltration of macrophages in STS [29]. As stimulation of anti-tumour lymphocytic mechanisms are critical to the success of immunotherapy, the lymphocyte-poor TIME of sarcomas poses a significant barrier to emerging immunotherapies. Thus, strategies capable of dismantling the immune quiescent sarcoma TIME are necessary to improve the efficacy of immunotherapies in STS.

Over the past two decades, numerous inducible, transgenic mouse models of bone and soft tissue sarcoma have been developed and characterized, however, many of these models are limited by prolonged tumour latency [30–33], multiple synchronous or metachronous tumours in diverse anatomic locations [34], and mixed genetic backgrounds [30, 33, 35], thus limiting transplantable, syngeneic applications. The cre-recombinase $Kras^{G12D};Trp53^{fl/fl}$ mutation (KP) model of UPS was first described by Kirsch et al. [36], and with increased interest in the immune landscape of human STS, characterizing the immunobiology of this versatile KP UPS model is important as we utilize this model for pre-clinical immunotherapy applications.

Here we describe our histologic, transcriptomic and immune characterization of inducible and transplanted KP UPS sarcomas in C57Bl/6 mice. Following intra-muscular lenti-cre recombinase injections into the muscular hindlimb, spontaneous tumours are > 90% penetrant with a short 2–3 month latency. Tumour histology and gene expression profiles of these murine UPS tumours matches human UPS. Cell lines from spontaneous tumours can be reproducibly orthotopically engrafted in the hindlimb and in the lung. Like many human STS, the TIME of both spontaneous and cell line derived murine UPS recapitulate the macrophage rich, immune quiescent phenotype common to most human STS subtypes. Despite increase lymphocytes in the transplanted tumours, like many human STS, this preclinical model was also resistant to immune checkpoint blockade.

## Methods

### Mouse strains

All animal procedures were performed with approval from the University of Calgary, Health Sciences Animal Care Committee (protocols AC15-0031 and AC19-0072). All mice were

housed in a biohazard level 2 biocontainment facility of the University of Calgary, Health Sciences Centre. Food and water were available to mice *ad libitum*. The animal facility was kept at a temperature of 22˚C, humidity of 30–35%, and a 12-hour light/dark cycle. B6.129P2-*Trp53^tm1Brn*/J (Trp53^fl) and B6.129S4-*Kras^tm4Tyj*/J (*Kras^G12D*) mice were obtained from The Jackson Laboratory (JAX; Stock #s: 008462 and 008179, respectively). Both mouse strains have been extensively backcrossed onto an inbred C57Bl/6 genetic background. The *Trp53^fl* mouse strain is hemizygous for Lox-P sites flanking exons 2 and 10 of the *Trp53* gene; the *Kras^G12D* mouse strain is hemizygous for a Lox-Stop-Lox (LSL) sequence upstream of a *Kras G12D* point mutation allele. *Trp53^fl/+* x *Kras^G12D/+* breeder pairs were used to generate *Trp53^fl/fl*; *Kras^G12D/+* offspring. All offspring were genotyped by polymerase chain reaction (PCR) for the presence of *Trp53^fl/fl* and *Kras^G12D/+* alleles using gene-specific primers [37, 38]. All mice who received subperiosteal injections of lenti-cre were given 100mg/kg of ketamine and 10mg/kg xylazine for comfort and conscious sedation throughout the procedure. Control C57Bl/6 (n = 21), Rag2 (n = 9) knockout mice (Rag2 KO, B6(Cg)-*Rag2^tm1.1Cgn*/J), and C57Bl/6 (n = 16) mice used to assess immune checkpoint blockade therapy were also obtained from The Jackson Laboratory (Stock #s 000664 and 008449, respectively). For spontaneous tumour induction experiments, an equal distribution of 6-8-week-old male and female mice were used (n = 25).

Humane endpoint for all animals was defined as a tumour volume measurement that met or exceeded 15 millimeters in any dimension (length, depth, or width) as measured using digital calipers, or any observable measures of prolonged discomfort (unkept coat, weight loss, difficulty ambulating or lethargy). Tumour volume measurements and animal health observations were completed weekly and then advanced to three times a week once tumours were discovered. Once humane endpoint criteria were observed, mice were immediately euthanized via $CO_2$ inhalation. No mice were prematurely euthanized prior to the previously described humane endpoints. If no tumours were observed greater than six months from lenti-cre injection, mice were euthanized via $CO_2$ inhalation. All personnel performing experiments with and monitoring mice have successfully completed animal handling and surgical training through the Institutional Animal User Training Program (IAUTP).

## Spontaneous tumour induction

Lenti-virus assembled with a CMV-driven cre recombinase plasmid (lenti-cre) was obtained from the University of Iowa Viral Vector Core. Under sterile conditions, using a 26-gauge needle tip, the proximal-lateral periosteum of the tibia was excoriated 10 times to establish an injury and healing response which increases penetrance and shortens the latency of sarcoma development [39]. This was immediately followed by an injection of 10μl of $1 \times 10^8$ pfu lenti-cre into the deep margin on the tibialis anterior muscle of the right hindlimb in *Trp53^fl/fl*; *Kras^G12D/+* mice. Experiment endpoint was reached if a tumour exceeded 15mm (in the length, depth, or width dimensions), or if mice reached 12 months of age. For each mouse, a complete necropsy was performed in collaboration with a clinical sarcoma pathologist (D.I.). The injected hindlimb, draining iliac and lumbar lymph nodes, the liver, spleen, and both lungs were examined macroscopically and microscopically for evidence of neoplasia. In tumour-bearing mice, visibly viable tumour was equally partitioned for histology, cell line development, or snap frozen using liquid nitrogen and stored at -80˚C for prospective molecular analyses.

## Diagnostic histopathology

Tumours submitted for histopathology were fixed in 10% neutral buffered formalin (Research Products International Corp) overnight and subsequently transferred to 70% ethanol and

embedded in paraffin blocks. Tumour sections were cut to a thickness of five microns for all slide preparations. Hematoxylin and eosin (H&E) staining was performed on all tumour and lung specimens using standard clinical pathology protocols at our institute. The following antibodies (Dako) were used for diagnostic immunohistochemical phenotyping of tumours: monoclonal mouse anti-cytokeratin–clone AE1:3 (GA053), monoclonal mouse anti-smooth muscle actin–clone 14A (IR611), monoclonal mouse anti-desmin–clone D33 (IR606), monoclonal mouse anti-myogenin–clone F5D (M3559) and rabbit polyclonal anti-S100 (GA504). FFPE tissue sections (5μm) were cleared in xylene and rehydrated in serial concentrations of ethanol. After peroxidase blocking for five minutes, sections were incubated with horseradish peroxidase conjugated primary antibodies (1:100) for 15 minutes. Immunoperoxidase staining was performed according to the manufacturer's protocol. Photomicrographs were obtained on a Leica DM5500B microscope using Leica Application Suite software.

## RNA sequencing and comparative genomics

RNA from fresh spontaneous hindlimb sarcomas was isolated using the RNeasy MiniKit (Qiagen). RNA samples were prepared and sequenced by Novogene Corporation Ltd (Sacramento, CA) using an Illumina Hiseq4000 platform. RNAseq was performed using paired-end reads, 150bp for each read, with 42–60 million reads for each sample. Three histologically verified spontaneous mouse UPS and three control contralateral muscle specimens were submitted for RNA sequencing. RNAseq files containing transcripts per million (TPM) listed as mRNA refseq IDs were converted into gene symbols using the biomaRt Bioconductor package. RNAseq files were uploaded to GEO (GSE174540). Heatmaps and PCA plots were created for the tumour and control muscle gene expression data sets to confirm appropriate gene clustering within tumours and skeletal muscle.

The publicly available human sarcoma RNAseq datasets (TCGA SARC [25], GSE75885 [40], GSE108022 [41])were utilized for comparative genomics. We limited our datasets to include skeletal muscle and five muscle lineage sarcoma subtypes relevant to our analysis: synovial sarcoma, undifferentiated pleomorphic sarcoma (UPS), alveolar rhabdomyosarcoma (ARMS), embryonal rhabdomyosarcoma (ERMS), and pleomorphic rhabdomyosarcoma (PRMS). We defined UPS in the TCGA dataset as UPS + myxofibrosarcoma (MFS), it has been shown in a comprehensive review of adult STS that UPS and MFS are indistinguishable across multiple sequencing platforms [25]. Only primary tumours were analyzed. For RNAseq datasets, Bioconductor SVASeq [42] was used to improve sample clustering and mitigate batch effects between the datasets. The biomaRt package was used to convert any mouse-specific homologs into their human gene symbol equivalent. Heat maps and dendrograms were created using unsupervised hierarchical clustering. RNAseq data was used to evaluate gene set enrichment (GSEA) of our tumour and control samples against reference datasets. We utilized the SAMSeq method [43] of the SAMR Bioconductor package to calculate the fold increase of each gene of the mouse model RNAseq data in tumour samples vs. skeletal muscle samples. The final gene set included genes with a minimum tenfold increase expression relative to muscle. Each gene set was then analyzed for enrichment in each sarcoma subtype vs. all other classifications. Pearson correlations were performed comparing the mouse tumour datasets with each reference dataset.

## Cell lines and syngeneic tumour model

Spontaneous tumours with a confirmed histologic UPS diagnosis were disaggregated and cultured in RPMI 1640 media (Gibco) with 10% Fetal Bovine Serum (FBS) (Gibco) in addition to 1% Penicillin and Streptomycin (PenStrep) (Gibco) and incubated at 37°C in 5% $CO_2$. The cell

line was passaged for four weeks and tumour cells (referred to herein as "TAO1") were lentivirus transduced with a mCherry-luciferase dual reporter vector (Addgene). Transduced cells were than FACS sorted to purify polyclonal population of mCherry positive cells. The M3-9-M was obtained from Crystal MacKall (Stanford, CA, USA) and was cultured in RPMI 1640 media (Gibco) with 10% FBS (Gibco) and incubated at 37˚C in 5% $CO_2$. Cell lines were tested for mycoplasma, and multiple mycoplasma negative stocks were frozen down for future *in vivo* experiments. Karyotyping and chromosome counts were performed on early and late passage UPS cell lines that were generated from spontaneous UPS tumours. The UPS cell lines were cultured to 90% confluency in 35-millimeter cell culture plates. Subsequently, the cells were trypsinized, fixed, mounted on slides, and prepared for karyotyping according to a previously published protocol [44].

For syngeneic hindlimb tumour development, 100,000 Luc+ TAO1 cells were injected into the tibialis anterior muscle of the right hindlimb in control C57Bl/6 or Rag2 KO mice. Tumor volume was calculated using the following equation: (L+X) (L) (X) (0.2618) [45]. The variable X is defined as the length of the tumour added to the width of the tumour divided by two (X = (L+W)/2), where length represents the largest tumor diameter and width represents the perpendicular tumor diameter. Tumour bioluminescence was measured (Xenogen, IVIS, Caliper Life Sciences) following a 150mg/kg of body weight intra-peritoneal (IP) injection of D-Luciferin (Sigma) reconstituted in PBS and reported as photon flux/sec. Bioluminescence imaging [46] was conducted 10 minutes following IP injection. Palpable and BLI detectable tumours are monitored weekly using BLI and volume measurements. For syngeneic lung metastases development, 100,000 Luc+ TAO1 cells were injected via the dorsal tail vein. BLI detectable lung signal was measured weekly until humane endpoints were reached.

## Flow cytometry

Tumours were excised and homogenized using a gentleMACS Dissociator (Miltenyi). Tumours were digested in RPMI media containing DNAse I (0.5mg/10mL, Sigma), Collagenase II (20mg/10mL, Sigma), and FBS (0.5mL/10mL, Gibco). Tumours were further processed for flow cytometry through straining with a 70μm strainer, treated with 10% RBC lysis buffer, and separated on a 80%-40% percoll gradient. Single cell suspensions were first stained with LIVE/DEAD Zombie Aqua (BioLegend, 423101). Subsequent antibody staining was completed using the following anti-mouse fluorophore-conjugated antibodies (BioLegend): CD3 (155610), CD4 (100512), CD8 (100733), PD-1 (135223), CD11c (117314), F4/80 (123122), CD45 (103154), CD11b (101207), and Ly6G (127615). Data was acquired using an Attune NXT flow cytometer (Invitrogen). The data was analyzed with FlowJo (TreeStar). Tumour infiltrating lymphocytes (TILs) are defined as and CD3+/CD8+/CD19- (CD8 TILs) and CD3+/CD4+/CD19- (CD4 TILs). Other immune populations of interest include macrophages (CD11b$^+$/Ly6G$^{low}$F4/80$^+$), neutrophils (CD11b$^+$/Ly6G$^{high}$), and dendritic cells (CD11b$^+$/CD11c$^+$). Spleens were harvested and processed into single cell suspensions and used as positive controls. Other controls included a dead cell sample, unstained cells, and single colour controls.

## Immunological checkpoint blockade therapy

100,000 Luc+ TAO1 cells were injected into the tibialis anterior muscle of the right hindlimb in control C57Bl/6 mice. Mice were treated with a mouse anti-CTLA4 monoclonal antibody (BioXcell, 250μg) or a mouse anti-PD1 monoclonal antibody (BioXcell, 250μg) IP, on days 7, 10, and 13 following TAO1 injection. Doses and schedules are consistent with other pre-clinical checkpoint inhibitor studies [47, 48]. Experimental endpoint was defined as maximum tumour diameter of 15mm in any direction.

## NanoString® immunological profiling

RNA was isolated from snap frozen samples of syngeneic and spontaneous tumours collected at humane endpoint. Immune-focused gene expression was quantified on 100ng of RNA using the NanoString® PanCancer Immune Profiling Panel assay (NanoString, Seattle, WA). Gene expression, immune pathway analyses, and statistical analyses were completed using nSolver advanced analysis software (version 4.0).

## Results

### Intramuscular *Trp53* and *Kras*$^{G12D}$ mutation induces rapid, localized high-grade myogenic sarcomas

The cre-mediated *Trp53* deletion and oncogenic *Kras*$^{G12D}$ activation results in a localized hindlimb spindle cell sarcoma with high penetrance and short latency (Fig 1). Following lenti-cre injections, spontaneous tumours developed within 2–3 months (range: 54–155 days, median: 79 days) with a penetrance of approximately 90% with 22 of 25 mice successfully developing hindlimb tumours (Fig 1A–1C). On routine H&E histology, spontaneous tumours display morphological features consistent with a high-grade, pleomorphic spindle cell sarcoma (Fig 1D and 1E). Immunohistochemical studies demonstrate strong actin staining and patchy Myogenin D expression (Fig 1F and 1G) indicative of poorly differentiated muscle lineage, consistent with a diagnosis of UPS. In this model, once hindlimb tumours are detected, all animals reached humane endpoint prior to the development of any gross or microscopic lung metastases.

### Gene expression profiles of murine hindlimb sarcomas resembles human UPS

To determine if the molecular signature of murine UPS tumours resembles that of human UPS, the gene expression profiles of *Kras*$^{G12D}$*Trp53* mutated murine sarcomas (n = 3) were

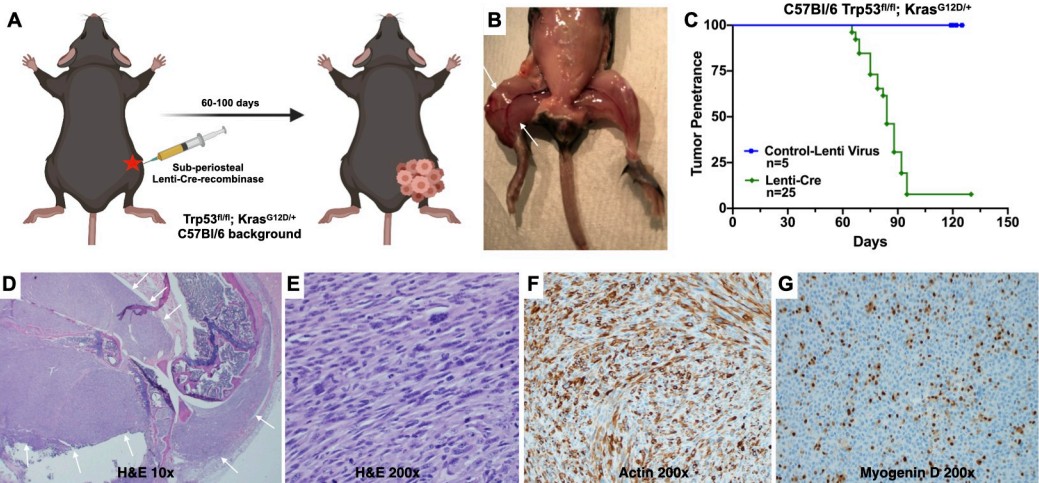

**Fig 1. Mutation of *Trp53* and *Kras* in C57Bl/6 mice results in rapid tumour formation.** (A-B) Sub-periosteal lenti-cre injection into the tibialis anterior muscle of C57Bl/6 mice results localized tumour growth involving the entire hindlimb. (C) Tumours develop in greater than 90% of injected mice with a short latency of 2–3 months from lenti-virus injection until tumours are palpable. (D-E)10x and 200x magnification photomicrographs of UPS tumours showing a malignant, highly pleomorphic spindle cell sarcoma. (F-G) Immunohistochemical studies show strong actin staining and patchy Myogenin expression, consistent with a diagnosis of UPS. White arrows delineate the tumour margin on both gross necropsy and low magnification microscopy.

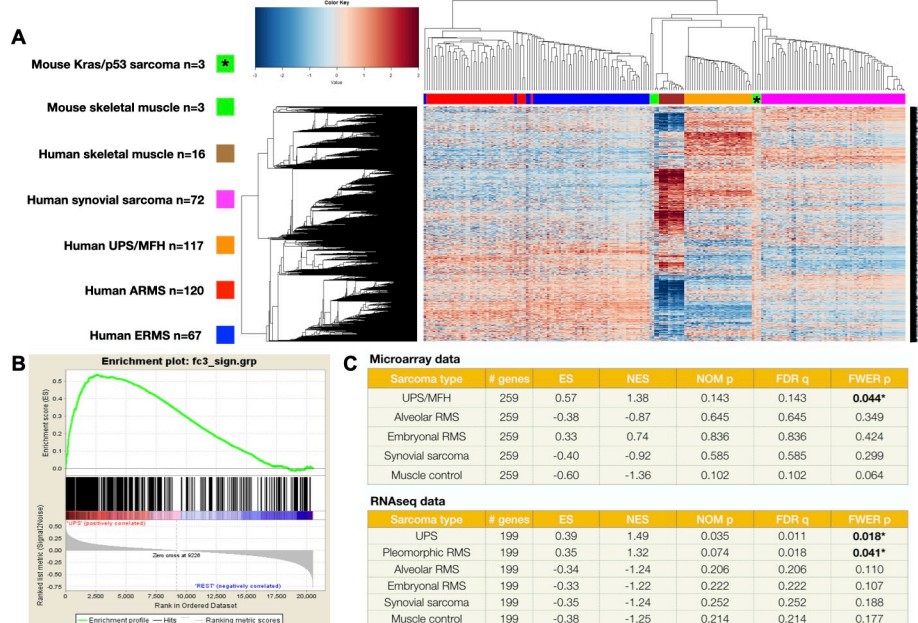

**Fig 2. Transcriptomic analyses show murine sarcomas resemble human UPS.** (A) Unsupervised hierarchical clustering of murine sarcoma RNAseq data with that of muscle lineage human soft tissue sarcoma shows murine sarcomas group with human UPS. (B-C) Additional gene set enrichment analyses the top differentially expressed genes of spontaneous murine tumours sarcomas is most similar to human UPS.

directly compared to common skeletal muscle lineage human sarcoma subtypes using publicly available RNAseq datasets (TCGA SARC [25], GSE75885 [40], GSE108022 [41]). Following normalization and mitigation of batch effects, unsupervised hierarchical clustering of murine tumours, murine control muscle, human myogenic-sarcomas and human control muscle was performed. As demonstrated in Fig 2A, RNAseq data of the spontaneous murine UPS tumours clustered with human UPS. Murine muscle RNAseq data also clustered with human muscle, further validating our data set. Additionally, using gene set enrichment analyses (GSEA), gene expression of the top ~200 differentially expressed genes in the spontaneous murine sarcomas was also most similar to human UPS (Fig 2B and 2C). Comparing the differential expression of mouse vs. human UPS, of 14,345 orthologous genes, 1,287 and 1,521 genes were differentially upregulated and downregulated, respectively. Gene ontology pathway analyses (g:Profiler) of differentially expressed murine genes relative to human UPS are listed in S1 Fig, S1 File, and S2 File. Collectively, these data support our histopathologic observations and provide transcriptomic evidence that these murine STS recapitulate the features of human UPS.

## Cell lines derived from spontaneous murine UPS tumours grow avidly in syngeneic C57Bl/6 mice and retain histologic features of UPS

Cell line derived, immunologically competent sarcoma models are scarce, yet critical to cancer immunology research. We thus sought to determine if cell lines from spontaneous UPS tumours could be transplanted into naïve, immunologically competent C57Bl/6 mice. Single cell suspensions of a disaggregated spontaneous UPS tumour from a female mouse were serial passaged to create a murine UPS cell line. G-banding and chromosome counts were completed using an early passage (P2) and a late passage of the TAO1 murine UPS (P25) and were found to retain similar amounts of chromosomes per cell, with a mean of 85 chromosome counts in

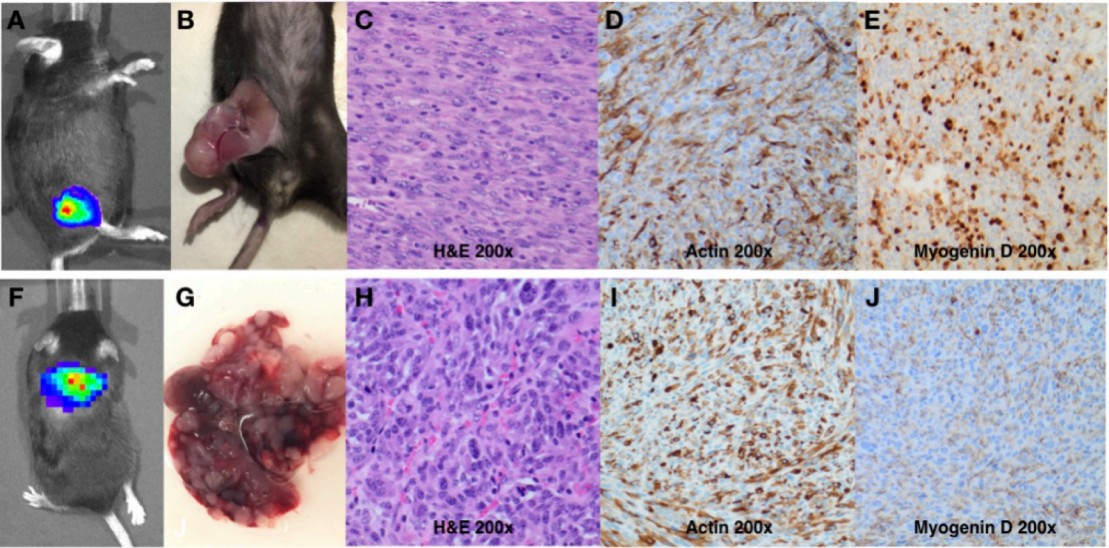

**Fig 3. Cell lines derived from spontaneous UPS tumours are easily engrafted in the hindlimb and lung of naïve, immunologically competent C57Bl/6 mice.** (A-B) Following injection of 100,000 UPS cells into the hindlimb of C57Bl/6 mice, tumours are detectable via bioluminescence imaging and result in large palpable tumours by three weeks. (C-E) Routine histology and immunohistochemistry of cell line derived tumours is consistent with UPS and similar to spontaneous UPS tumours. (F-G) Tail vein injections result in bioluminescence detectable tumours in the lung with multiple tumour lung nodules on gross examination of lung tissue. (H-J) Routine histology and immunohistochemistry of cell line derived lung tumours is consistent with UPS and similar to spontaneous UPS tumours.

the early passage and a mean of 88 chromosomes in the late passage (S2 Fig). Both cell lines are highly aneuploid with many copy number alterations. The early passage cell line has fewer copy numbers (17) compared to the later passage of the cell line (21). Once the line was established ("TAO1"), it was transduced with an mCherry-luciferase dual reporter vector and then FACS sorted to purify polyclonal populations. Following this, C57Bl/6 mice were given a single hindlimb injection of 100,000 TAO1 cells and monitored via BLI and tumour volume measurements to monitor tumour growth kinetics (Fig 3A and 3B). All mice injected with TAO1 developed tumours within 13–15 days.

H&E staining of transplanted UPS tumours share similar morphological characteristics with the spontaneous UPS tumours (Fig 3C). Immunohistochemical staining yielded comparable results to spontaneous UPS tumours showing strong positive skeletal actin and patchy Myogenin staining, indicating the UPS phenotype expressed in spontaneous tumours is retained in this syngeneic UPS cell line model (Fig 3D and 3E). At humane endpoint of transplanted UPS tumour experiments, no visible or microscopic lung metastases were observed. To evaluate if murine UPS cells develop tumour in the lung, 100,000 TAO1 UPS cells were transplanted via dorsal tail vein injection. All mice who received tail vein injections of TAO1 UPS cells developed BLI detectable UPS tumours in the lung (Fig 3F and 3G). Within 35 days of TAO1 UPS cell injection, all mice reached humane endpoint. All lung tumours retained the same histological characteristics of hindlimb tumours, consistent with a diagnosis of UPS (Fig 3H and 3I).

## Murine UPS tumours are characterized by a paucity of lymphocytes and are rich in tumour macrophages

We next sought to evaluate the immune populations in both spontaneous and cell line derived (transplanted) murine UPS tumours using flow cytometry. Spontaneous and transplanted

UPS tumours (n = 6 per tumour type, for each experiment) were harvested at humane end-point and immune populations, were dissociated into single cell suspensions and quantified. Labelling and gating strategies were developed to quantify tumour infiltrating lymphocytes (TILs) (CD3/CD4/CD8/CD19) and myeloid lineage cells (CD45$^+$/CD11b$^+$; macrophages: CD11b$^+$/Ly6G$^{low}$F4/80$^+$; neutrophils: CD11b$^+$/Ly6G$^{hi}$; and dendritic cells CD11b$^+$/CD11c$^+$). The myeloid lineage was the dominant immune population in spontaneous UPS types (Fig 4A). Tumour associated macrophages (TAMs) comprised 14.3% (+/- 1.6%) of all viable cells, while tumour infiltrating lymphocytes (TILs) were scarce; CD4$^+$ and CD8$^+$ TILs accounted for 1.4% (+/- 0.7%) and 1.8% (+/- 0.2%), respectively.

This observed paucity of TILs was further evaluated in comparison to an immunologically competent model of embryonal rhabdomyosarcoma (ERMS), M3-9-M, a cell line derived from an ERMS tumour that arose in a male C57Bl/6 $HGFT^{+/-};Trp53^{+/-}$ mouse [49]. The M3-9-M line is immunogenic with pronounced lymphocyte infiltrates when transplanted in female mice (M3-9-M-F), but less immunogenic in males (M3-9-M-M), owing to recognition of male HY antigens by female T cells [49, 50]. Both CD4+ and CD8+ TILs proportions in spontaneous and transplanted UPS tumours were low: CD4+ 0.8% +/- 0.2 and 1.8% +/- 0.4, respectively; CD8+ 1.1% +/- 0.4 and 3.7% +/- 2.0, respectively (Fig 4B). These numbers were significantly less than CD4+ and CD8+ populations in M3-9-M-Female tumours (p<0.001, one-way ANOVA with post-hoc Tukey). We observed roughly double CD4+ and triple the CD8+ lymphocytic infiltrates in transplanted vs. spontaneous UPS tumours (Fig 4B), but these differences were not statistically significant (p = 0.48 and p = 0.20, respectively). In these

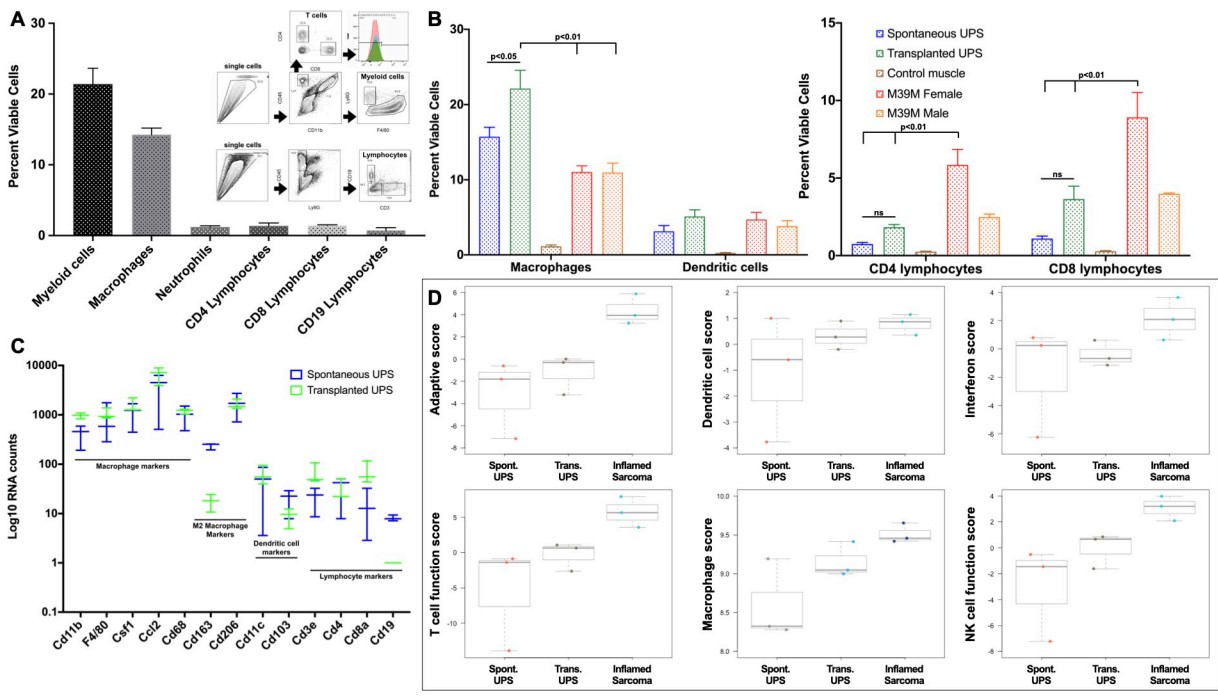

**Fig 4. Spontaneous and cell line derived murine UPS tumours are macrophage rich and lymphocyte poor.** (A) Using FACS analyses, macrophages are the dominant intra-tumoural immune population (14%), whereas lymphocyte numbers are low (<2%). (B) In a separate analysis comparing spontaneous UPS, cell lined derived (transplanted) UPS and the inflamed M3-9-M ERMS model, CD4 and CD8 lymphocyres are low (1–4%) and significantly less than the infamed M3-9-M-Female model (6–9%). The cell line UPS model is more inflamed and contains more lymphocytes than the spontaneous model, althogh these differences did not reach stastistical significance. (C-D) NanoString® immune profiling of UPS tumours also demonstrates high macrophage scores, and negative lymphocyte, NK cell, dendritic cell and adaptive immune pathway scores compared to an inflamed sarcoma model.

experiments, TAMs were the dominant immune population in both spontaneous (16% +/- 3.2) and transplanted models (22% +/- 6.0) (Fig 4A and 4B). More TAMs were observed in transplanted model, which was significant (p = 0.02, one-way ANOVA with post-hoc Tukey).

In a complimentary series of experiments, we evaluated the immune transcriptional signatures of spontaneous UPS (n = 3), transplanted UPS (n = 3), and transplanted UPS treated with an intra-tumoural dose of a stimulator of interferon genes (STING) agonist, DMXAA (18mg/kg, Sigma) ('inflamed sarcoma', n = 3) using the NanoString® Pan Cancer Immune Panel. mRNA counts of macrophage markers were high, while lymphocyte markers were low in both spontaneous and transplanted UPS (Fig 4C). Interestingly, CD163, a commonly used marker of M2 polarized macrophages [28, 29] was significantly higher in spontaneous UPS tumours compared to transplanted UPS tumours (p<0.001, one-way ANOVA with post-hoc Tukey). Utilizing the NanoString® nSolver advanced analysis tool, immune pathway scores as a function of sarcoma sub-type are generated from individual transcript numbers. A negative score is indicative of pathway downregulation, and a positive score indicates upregulation of that immune pathway. In both spontaneous and transplanted UPS models, we see negative immune scores for adaptive immunity, T-cell functions, interferon signaling, natural killer (NK) cell functions, and very positive macrophage scores (Fig 4D). These data are consistent with immune phenotype data from flow cytometry experiments showing a paucity of lymphocytes and adaptive immunity and enrichment of TAMs in both spontaneous and transplanted UPS models.

## In vivo UPS growth kinetics are unaffected by lymphocyte deficiency and UPS tumours are resistant to immune checkpoint inhibitors

Despite low levels of spontaneous lymphocytic infiltrates in this UPS model we sought to evaluate the impact of lymphocyte contributions towards tumour growth parameters. To accomplish this, 100,000 UPS cells were transplanted orthotopically into the hindlimb of lymphocyte deficient Rag2 KO mice. Compared to immunologically competent C57Bl/6 mice, tumours transplanted in Rag2 KO mice displayed similar tumour growth kinetics and survival time (Fig 5A and 5B). These

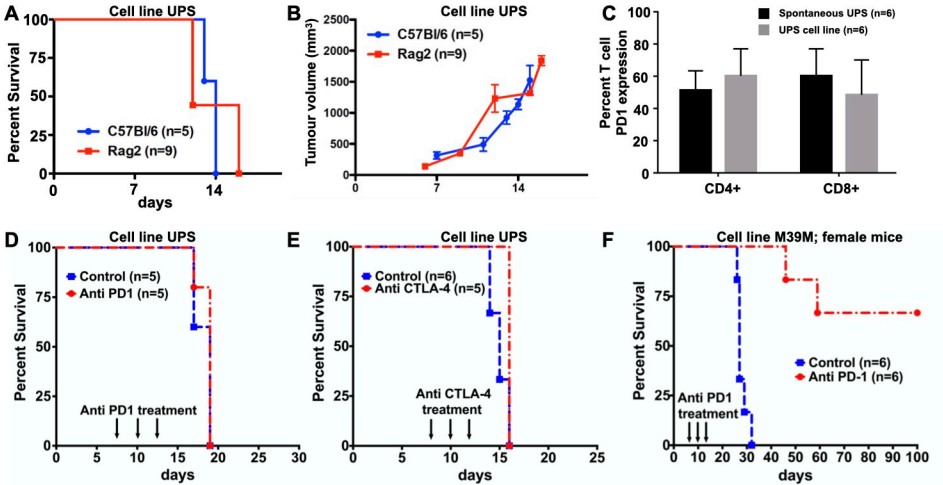

**Fig 5. Murine UPS tumour growth is unaffected by lymphocyte deficiency and resistant to immunological checkpoint blockade therapy.** (A-B) Survival and growth kinetics of UPS cells engrafted into lymphocyte deficient Rag2 KO mice were no different from control C57Bl/6 mice. (C) Despite 45–55% of CD4 and CD8 TILs expressing PD1, UPS bearing mice were resistant to anti-PD1 and anti-CTLA-4 therapy, whereas survival was 66% in the inflamed M3-9-M model similarly treated with anti-PD1 (D-F).

results show that transplanted UPS tumour growth in this model is not measurably influenced by spontaneous lymphocytic infiltrates.

Next, we investigated if UPS tumours were sensitive to both anti-PD1 and anti-CTLA-4 immune checkpoint blockade. Data from flow cytometry experiments shows that 45–55% of UPS CD4 and CD8 TILs expressing PD1 (Fig 5C), yet when UPS bearing mice are treated with either anti-CTLA4 or anti-PD1, no changes in survival were observed (Fig 5D and 5E). The immunogenic M3-9-M ERMS model, which is enriched in TILs was sensitive to anti-PD1 therapy with 66% overall survival beyond three months (Fig 5F). These results demonstrate this lymphocyte poor UPS model is resistant to conventional immunological checkpoint inhibitors and recapitulates the therapeutic insensitivity observed in recent clinical trials evaluating immunological checkpoint inhibition across a variety of STS subtypes.

## Discussion

Inducible and syngeneic mouse models will be important tools in the advancement of sarcoma immunology and immunotherapies. Inducible models develop in a more natural host environment and are not influenced by selective pressures of tissue culture, although the longer latency and variable timelines for tumour develop limit the versatility of these models in larger scale experiments. Transplantable syngeneic models are much easier to experimentally manipulate and control, yet are limited by clonality and a more unnatural process of tumour formation. Using orthotopic hindlimb mutations of $Trp53^{fl/fl}$ and $Kras^{G12D}$ in iC57Bl/6 mice, we induced spontaneous sarcomas that are histologically and transcriptomically similar to human UPS. This mutational strategy is aligned with the molecular fingerprints associated with clinical sarcomas: Mutations in the *p53* gene are very common in human sarcomas and are detected in 50–70% of STS [25]; and although allelic point mutations of *RAS* are less common in STS, more than 80% of human UPS display aberrant activation of the RAS/MAPK pathway [51, 52]. Notably, we have also generated cell lines derived from spontaneous tumours that retain UPS histology and grow easily in syngeneic, immunologically competent C57Bl/6 mice. Although we did not observe macroscopic or microscopic lung metastases in spontaneous or transplanted models, tail vein injections of UPS cells consistently resulted in UPS engraftment within the lung. Furthermore, we have also shown that immune microenvironment in both spontaneous and transplantable UPS models here are lymphocyte poor, rich in TAMs and express low transcriptome pathway scores for adaptive immunological functions. This phenotype is also consistent with immunological profile of most human STS [12, 20, 22, 23, 25, 26, 29].

Our study demonstrates that the KP murine model of UPS harbors a TIME lacking spontaneous adaptive immunity. Other groups have similarly documented the immunologically 'cold' dominant human sarcoma phenotype, characterized by a paucity of tumour lymphocytes [21–23], macrophage enrichment, and immunologically suppressive signaling in the TIME [24–29]. The precise factors responsible for the immunologically quiescent sarcoma TIME remain poorly understood although can likely be attributed to a combination of a low mutational in sarcomas relative to other cancers [53, 54], an immunologically suppressive connective tissue signature [24] and potentially the influence of increased M2 polarized macrophage densities in sarcoma [29]. Indeed, the mutational burden of $Kras^{G12D/+};Trp53^{fl/fl}$ induced murine sarcomas are low with less than 0.1 mutations/Mb [55]. As there are numerous histologically and molecularly distinct subtypes of STS, Petitprez et al. [20] recently used genomic data to classify sarcomas based on the tumour TIME (Sarcoma Immune Class, SIC), defining these as 'immunologically cold' (SIC A, SIC B), 'vascular' (SIC C) or 'immune high' (SIC D, SIC E). Immunologically cold (SIC A and B), the most common STS category, predicted poor

clinical response to immunologic checkpoint blockade. Human UPS is a heterogenous sarcoma subtype and unsurprisingly, in this study, UPS was represented in all sarcoma immune classes. In the SIC A and B sarcomas, immune cells represented < 30% of the total cell fraction, with macrophages representing the dominant immune population [20, 56]. The SIC A and B sarcomas were also associated with resistance to immune checkpoint inhibitors. In our experiments, the number of total immune cells in both spontaneous and transplanted KP UPS models were low (<30%), and tumours were resistant to anti-PD1 and anti-CTLA4, thus we believe the KP UPS model characterized in this study is most aligned with a SIC A or B immunologically cold phenotype. In a recent study by Wisdom et al. [56], they immunologically characterized a high mutation load mouse model of UPS (80-fold higher than the KP model) induced using a combination of *Trp53* and carcinogen 3-methylcholanthrene (MCA) mutagenesis. In this model of UPS, both spontaneous and transplanted sarcomas were consistent with a SIC D or E (immune high) phenotype and were therapeutically sensitively to immune checkpoint blockade. These findings highlight how different mutagenic approaches can be used to develop immunologically distinct murine UPS models that importantly, recapitulate the diversity of this disease observed in humans. Highlighting this, in the SARC028 phase 2 trial of single agent pembrolizumab for advanced sarcomas [10], four of ten patients with UPS demonstrated an objective response, likely owing to the immunologic heterogeneity within this STS subtype. Synthesizing this clinical and pre-clinical data, we believe the *Trp53/Kras* KP model of UPS and the *Trp53*/MCA model of UPS can be utilized to study immunologically cold vs. inflamed UPS subtypes.

Consistent with other studies in this field [56, 57], the TIME of transplanted KP tumours was more inflamed than spontaneous UPS, showing a 2-fold and 3-fold increase in CD4+ and CD8+ lymphocytes. These finding were also observed on a transcriptomic level as the Nano-String® immunological profiling also demonstrated a trend towards higher T cell, adaptive immunity and NK pathways scores in the transplanted model, however these differences were not statistically significant. In a recent study by Gutierrez et al. [57], examining spontaneous and transplanted *Trp53/Kras* models of UPS and ERMS, a 2–3 fold increase in CD4 and CD8 lymphocytes were also observed in both models. Similar results have been reported by Riddell et al., showing that spontaneous cre-induced *Trp53;Kras* KP lung cancer tumours are less immunogenic than transplanted KP lung models [58]. Despite the increased lymphocyte numbers we observed in transplanted KP UPS tumours, the *in vivo* growth parameters of transplanted tumours was unaffected by lymphocyte deficiency (Rag2 KO) and furthermore, tumours remained resistant to immunologic checkpoint blockade, suggesting persistence of the immunologically cold phenotype in a therapeutic context as well. Regardless, the immunologic differences between spontaneous and transplanted UPS tumours are important considerations when evaluating future immunologically-based therapeutic studies with this model. The biologic importance of this was highlighted by Wisdom et al., [56], as in the *Trp53*/MCA model of UPS, transplanted tumours were sensitive to immune checkpoint blockade, while spontaneous tumours were not.

Across multiple immune competent murine sarcoma models, macrophages are the most common immune cell of the TIME [56, 57, 59]. These findings are also observed in human sarcomas [20, 25, 29, 56, 60], although how this dominant myeloid compartment within sarcomas influences tumour immunobiology remains unclear. Recent clinical data from over 1200 STS patients reaffirms the rich, predominantly M2 infiltration of macrophages in sarcoma [29]. It is hypothesized that the anti-inflammatory, pro-tumour M2 macrophage population may contribute to an immune quiescent TIME. In our spontaneous KP tumours, we observed increased expression of the M2 marker, CD163, relative to the transplanted tumours, although the significance of this observation was not further assessed in this study. Certainly, given the

lack of lymphocytes, NK cells and dendritic cells in sarcomas, understanding how sarcoma TAMs can be manipulated or harnessed to induce a more immunogenic TIME is desirable and the KP UPS model would be an ideal system for these studies.

## Conclusion

Immunologically competent pre-clinical models of STS are critical to advance our understanding of sarcoma immunobiology and evaluate new immunologically based therapies. Our results demonstrate that both spontaneous and transplanted KP models of UPS recapitulate the lymphocyte poor, macrophage rich, immunologically cold TIME common to many human STS. This model represents an optimal pre-clinical tool for prospective research focused on discovering and evaluating new immunotherapeutic strategies for sarcoma patients.

## Supporting information

**S1 File. Murine UPS downregulated gene list.** Complete list of differentially downregulated murine genes relative to human UPS.
(PDF)

**S2 File. Murine UPS upregulated gene list.** Complete list of differentially upregulated murine genes relative to human UPS.
(PDF)

**S1 Fig.** Gene expression signature of murine UPS is similar to human UPS (A) Volcano plot showing the differentially expressed genes in murine UPS relative to human UPS with a log fold change of 2 and a p value set to 0.01. Of the 14,345 genes analyzed, 11,537 genes (80%) were not differentially expressed demonstrating the similarities between the murine and human UPS. (B) Biological pathway analysis of the 1,281 genes (9%) that were downregulated in murine UPS relative to human UPS showing the top ten pathways associated with the downregulation. (C) Biological pathway analysis of the 1,521 upregulated genes (11%) in murine UPS relative to human UPS showing the top ten pathways associated with the upregulation.
(TIF)

**S2 Fig. Chromosome copy number and aneuploidy remains consistent between early and late passage TAO1 UPS cell lines.** (A) Graphical representation of chromosome copy numbers for early and late passage TAO1 UPS cell lines as a function of normal diploid mouse chromosome count which is 40. 1x is defined as any chromosome count between 40–79, 2x as 80–119, 3x as 120–159, and 4x as 160–200. (B) G-banding of two cells with 45 chromosomes for both the early and late passage TAO1 UPS cell lines.
(TIF)

## Author Contributions

**Conceptualization:** Douglas J. Mahoney, Frank R. Jirik, Michael J. Monument.

**Data curation:** Reid McNeil, Franz Zemp, Jahanara Rajwani, Doha Itani, Pinaki Bose, Douglas J. Mahoney.

**Formal analysis:** Reid McNeil, Franz Zemp.

**Funding acquisition:** Michael J. Monument.

**Investigation:** Arvind K. Singla, Franz Zemp, Jahanara Rajwani.

**Methodology:** Arvind K. Singla, Doha Itani.

**Resources:** Frank R. Jirik, Michael J. Monument.

**Supervision:** Frank R. Jirik.

**Visualization:** Karys M. Hildebrand.

**Writing – original draft:** Karys M. Hildebrand, Kayla L. Marritt, Michael J. Monument.

**Writing – review & editing:** Kurt N. Hildebrand.

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
