## [Decision Letter · Decision Letter 0]

24 Mar 2021

PONE-D-21-03535

Characterization of a novel mouse model of undifferentiated pleomorphic sarcoma that is lymphocyte poor, macrophage rich, and resistant to immune checkpoint blockade

PLOS ONE

Dear Dr. Monument,

Thank you for submitting your manuscript to PLOS ONE. After careful consideration, we feel that it has merit but does not fully meet PLOS ONE’s publication criteria as it currently stands. Therefore, we invite you to submit a revised version of the manuscript that addresses the points raised during the review process.

Please give a spcific attention to the comments of reviewer#1 and the lacking references.

We look forward to receiving your revised manuscript.

Kind regards,

Dominique Heymann, Ph.D.

Academic Editor

PLOS ONE

Journal Requirements:

2. We note that you are reporting an analysis of a microarray, next-generation sequencing, or deep sequencing data set. PLOS requires that authors comply with field-specific standards for preparation, recording, and deposition of data in repositories appropriate to their field. Please upload these data to a stable, public repository (such as ArrayExpress, Gene Expression Omnibus (GEO), DNA Data Bank of Japan (DDBJ), NCBI GenBank, NCBI Sequence Read Archive, or EMBL Nucleotide Sequence Database (ENA)). In your revised cover letter, please provide the relevant accession numbers that may be used to access these data. For a full list of recommended repositories, see http://journals.plos.org/plosone/s/data-availability#loc-omics or http://journals.plos.org/plosone/s/data-availability#loc-sequencing.

Reviewers' comments:

Reviewer's Responses to Questions

**Comments to the Author**

1. Is the manuscript technically sound, and do the data support the conclusions?

Reviewer #1: Partly

Reviewer #2: Yes

2. Has the statistical analysis been performed appropriately and rigorously? 

Reviewer #1: Yes

Reviewer #2: Yes

3. Have the authors made all data underlying the findings in their manuscript fully available?

Reviewer #1: Yes

Reviewer #2: No

4. Is the manuscript presented in an intelligible fashion and written in standard English?

Reviewer #1: Yes

Reviewer #2: Yes

5. Review Comments to the Author

Reviewer #1: This article completely ignores over a decade of work that has been performed within the field of sarcoma using almost identical models. It makes multiple false claims, including that it is the first to describe syngeneic applications of murine UPS cells derived from KP models, and that there have been no immune profiling done of these models. A few citations describing immune profiling of models and syngeneic applications include: (these are by no means inclusive): PMID: 33335088, 31980651, 26398162, 26234681, 27035817, 33441747, 28841687). Therefore, this manuscript represents only a small, incremental step in the field and does not merit publication as it is currently written.

Reviewer #2: The article "Characterization of a novel mouse model of undifferentiated pleomorphic sarcoma that

is lymphocyte poor, macrophage rich, and resistant to immune checkpoint blockade" by Michael J. Monument and colleagues is suitable for publication in Plos One with minor revisions. They present a well conducted and controlled series of experiments characterizing the 'KP' model of soft tissue of sarcoma originally developed by the Tyler Jacks laboratory and new cell line derived from this model in terms of its immune cell repertoire and response to anti-CTLA4 and PD-1 therapy.

Before final publication I recommend the following changes are made

1. The authors need to clarify their definition of UPS. The TCGA SARC dataset has separate histotypes for UPS and Myxofibrosarcoma (MXF previously MFH). Have the authors excluded MXF, in which case this should be included, or have they combined this with the UPS histotype as sometimes happens in other studies? This needs clarification.

2. Although the hierarchical clustering shows the Cre induced sarcoma model is closest to UPS of the muscle derived STS examined, the heat map clearly shows genes which are both increased and decreased with respect to the KP model of STS. Differential gene expression between clinical UPS and the mouse model should be shown and discussed so readers can be clear what the main differences are when considering the use of this model.

3. The authors should provide karyotyping information for the new cell line and compare this to cells from the tumours as part of their discussion of how similar the cell line is to the cells within the tumour.

4. The RNAseq data (preferably FASTQ files), GESA, differential gene expression etc should be made publicly available, as per the policy of PLoS One.

6. PLOS authors have the option to publish the peer review history of their article (what does this mean?). If published, this will include your full peer review and any attached files.

Reviewer #1: No

Reviewer #2: No

---

## [Author Response · Author response to Decision Letter 0]

21 May 2021

Response to reviewers

Reviewer #1: This article completely ignores over a decade of work that has been performed within the field of sarcoma using almost identical models. It makes multiple false claims, including that it is the first to describe syngeneic applications of murine UPS cells derived from KP models, and that there have been no immune profiling done of these models. A few citations describing immune profiling of models and syngeneic applications include: (these are by no means inclusive): PMID: 33335088, 31980651, 26398162, 26234681, 27035817, 33441747, 28841687). Therefore, this manuscript represents only a small, incremental step in the field and does not merit publication as it is currently written.

We apologize for this oversight and misunderstanding. We have significantly revised the introduction and discussion to highlight the important scientific contributions towards this topic by the Kirsch lab and others. We have provided additional detail and discussion of how our present study reaffirms findings and conclusion from previous work. This has enabled us a more succinct opportunity to highlight the key messages of our work and convey how this study furthers our understanding of sarcoma models and sarcoma immunology. 

Reviewer #2: The article "Characterization of a novel mouse model of undifferentiated pleomorphic sarcoma that is lymphocyte poor, macrophage rich, and resistant to immune checkpoint blockade" by Michael J. Monument and colleagues is suitable for publication in Plos One with minor revisions. They present a well conducted and controlled series of experiments characterizing the 'KP' model of soft tissue of sarcoma originally developed by the Tyler Jacks laboratory and new cell line derived from this model in terms of its immune cell repertoire and response to anti-CTLA4 and PD-1 therapy.

Before final publication I recommend the following changes are made

1. The authors need to clarify their definition of UPS. The TCGA SARC dataset has separate histotypes for UPS and Myxofibrosarcoma (MXF previously MFH). Have the authors excluded MXF, in which case this should be included, or have they combined this with the UPS histotype as sometimes happens in other studies? This needs clarification.

Thank you for your review and suggestions.

We included myxofibrosarcoma and UPS in the TCGA SARC dataset used in our comparative approach. We have specified this in the manuscript (page 8, lines 436-438) and have highlighted the results from a 2017 Cell publication from TCGA Research Network demonstrating the largely indistinguishable genomics and transcriptomic profiles of UPAS and MFS.

2. Although the hierarchical clustering shows the Cre induced sarcoma model is closest to UPS of the muscle derived STS examined, the heat map clearly shows genes which are both increased and decreased with respect to the KP model of STS. Differential gene expression between clinical UPS and the mouse model should be shown and discussed so readers can be clear what the main differences are when considering the use of this model.

We tabulated the ~14,000 overlapping orthologous genes profiled in our comparative transcriptome analysis between human and mouse UPS (page 13, lines 562-567; S1Fig). We have also included differential expression figures, gene ontology analyses and .csv files listing all genes and associated differential expression (S1A & S1B File and S1.Fig).

3. The authors should provide karyotyping information for the new cell line and compare this to cells from the tumours as part of their discussion of how similar the cell line is to the cells within the tumour.

We have added G-banding and chromosomal analyses from early (P2) and late passage (P25) TAO1 cell lines (page 14, lines 601-606; S2. Fig). We were unable to obtain banding and chromosomal analyses for primary tumours using the available core services at our institute. Commercial karyotyping was either cost or time prohibitive for this revision.

4. The RNAseq data (preferably FASTQ files), GESA, differential gene expression etc should be made publicly available, as per the policy of PLoS One.

We apologize for this oversight. We have uploaded our RNAseq data in the appropriate file formats to GEO (GSE174540).

---

## [Editor Report · Decision Letter 1]

15 Jun 2021

The KrasG12D;Trp53fl/fl Murine Model of Undifferentiated Pleomorphic Sarcoma is Macrophage Dense, Lymphocyte Poor, and Resistant to Immune Checkpoint Blockade.

PONE-D-21-03535R1

Dear Dr. Monument,

We’re pleased to inform you that your manuscript has been judged scientifically suitable for publication and will be formally accepted for publication once it meets all outstanding technical requirements.

Kind regards,

Dominique Heymann, Ph.D.

Academic Editor

PLOS ONE
---

## [Editor Report · Acceptance letter]

30 Jun 2021

PONE-D-21-03535R1 

The *Kras^G12D^*;*Trp53^fl/fl^* Murine Model of Undifferentiated Pleomorphic Sarcoma is Macrophage Dense, Lymphocyte Poor, and Resistant to Immune Checkpoint Blockade. 

Dear Dr. Monument:

I'm pleased to inform you that your manuscript has been deemed suitable for publication in PLOS ONE. Congratulations! Your manuscript is now with our production department. 

Kind regards, 

on behalf of

Pr. Dominique Heymann 

Academic Editor

PLOS ONE